# A mechanistic investigation of the Li$_{10}$GeP$_2$S$_{12}$|LiNi$_{1-x-y}$Co$_x$Mn$_y$O$_2$ interface stability in all-solid-state lithium batteries

Tong-Tong Zuo [1,2✉], Raffael Rueß[1,2], Ruijun Pan[1,2,3], Felix Walther[1,2], Marcus Rohnke [1,2], Satoshi Hori[4], Ryoji Kanno[4], Daniel Schröder [5✉] & Jürgen Janek [1,2✉]

All-solid-state batteries are intensively investigated, although their performance is not yet satisfactory for large-scale applications. In this context, the combination of Li$_{10}$GeP$_2$S$_{12}$ solid electrolyte and LiNi$_{1-x-y}$Co$_x$Mn$_y$O$_2$ positive electrode active materials is considered promising despite the yet unsatisfactory battery performance induced by the thermodynamically unstable electrode|electrolyte interface. Here, we report electrochemical and spectrometric studies to monitor the interface evolution during cycling and understand the reactivity and degradation kinetics. We found that the Wagner-type model for diffusion-controlled reactions describes the degradation kinetics very well, suggesting that electronic transport limits the growth of the degradation layer formed at the electrode|electrolyte interface. Furthermore, we demonstrate that the rate of interfacial degradation increases with the state of charge and the presence of two oxidation mechanisms at medium (3.7 V $vs.$ Li$^+$/Li < $E$ < 4.2 V $vs.$ Li$^+$/Li) and high ($E \geq$ 4.2 V $vs.$ Li$^+$/Li) potentials. A high state of charge (>80%) triggers the structural instability and oxygen release at the positive electrode and leads to more severe degradation.

[1] Institute of Physical Chemistry, Justus Liebig University Giessen, Heinrich-Buff-Ring 17, D-35392 Giessen, Germany. [2] Center for Materials Research (LaMa), Justus-Liebig-University Giessen, Heinrich-Buff-Ring 16, D-35392 Giessen, Germany. [3] Materials Science and Engineering Program & Texas Materials Institute, The University of Texas at Austin, Austin, TX 78712, USA. [4] Institute of Innovative Research (IIR), All-Solid-State Battery Unit, Tokyo Institute of Technology, 4259 Nagatsuta, Midori-ku, Yokohama 226-8502, Japan. [5] Institute of Energy and Process Systems Engineering (InES), Technische Universität Braunschweig, 38106 Braunschweig, Germany. ✉email: tong-tong.zuo@pc.jlug.de; d.schroeder@tu-braunschweig.de; juergen.janek@pc.jlug.de

High-energy density is one of the major targets for energy storage systems in portable electronic devices and electric vehicles[1,2]. Although the energy densities of lithium-ion batteries with non-aqueous liquid electrolytes can still be improved, safety issues due to the flammable electrolyte and the hope for the safe use of the lithium metal anode drive the development of solid-state batteries (SSBs). In fact, SSBs with solid electrolytes (SEs) show great potential in addressing safety concerns, further improving the energy density as well as power density[3–5]. However, the unsatisfactory performance of SSBs is still a great challenge on the way to their practical application and commercialization. Besides the issues with dendrite formation at the lithium metal anode, the interfacial degradation of SEs and cathode active materials (CAMs) currently limits the electro-chemical performance of SSBs[6–12].

In order to meet the requirements for high-energy-density and high power density, layered oxide compounds from the LiNi$_{1-x-y}$Co$_x$Mn$_y$O$_2$ (NCM) series of solid solutions, like LiNi$_{0.6}$Co$_{0.2}$Mn$_{0.2}$O$_2$ (NCM622), are currently regarded as the best-suited cathode materials owing to the advantages of high specific capacity, high cycling stability and low cost[13]. In addition, SEs with very high ionic conductivities in the order of $10^{-2}$ S cm$^{-1}$ are required for the construction of the cathode composite with sufficient specific energy and rate capability[14]. Li$_{10}$GeP$_2$S$_{12}$ (LGPS) and its analogs with 3D framework structure show a very high ionic conductivity of $\sim 10^{-2}$ S cm$^{-1}$, and are therefore pro-totype SEs in fundamental studies[15,16]. Unfortunately, as all thiophosphate SEs, LGPS exhibits a narrow thermodynamic sta-bility window, due to its reactivity toward both cathode and anode materials[17,18]. The instability of the Li|LGPS interface has already been well studied[19,20]. It was found that a mixed ionic-electronic conducting interphase forms at the Li|LGPS interface, which grows fast and leads to a significant impedance increase. Thus, LGPS cannot be applied without protection against the lithium metal anode. The experimentally observed interfacial decomposition shows a $\sqrt{t}$ dependence for diffusion control and is in good agreement with theoretical predictions[17], which pro-vides a deeper understanding of the stability of the anode|SE interface.

Although instability of thiophosphate SEs at the cathode side was pointed out by theoretical computation[21], the SE|CAM interface, in particular, in the case of LGPS, has not been extensively studied experimentally. Some of the authors investi-gated the LGPS|LiCoO$_2$ interface by electrochemical impedance spectroscopy (EIS) and X-ray photoelectron spectroscopy, which revealed a severe degradation at the LGPS|LiCoO$_2$ interface[22,23]. Upon charging, lithium is extracted from LGPS and the forma-tion of poorly ion-conducting Li$_2$P$_2$S$_6$ and S$^0$ species is favored[18]. More recently, Walther et al.[24] proposed a reaction scheme for the SE degradation at β-Li$_3$PS$_4$|NCM622 interfaces and described the decomposition reactions at β-Li$_3$PS$_4$|carbon interfaces in detail. According to this study, the formation of oxygenated phosphorous and sulfur species (i.e., phosphates/phosphites, sulfates/sulfites) dominates the interfacial reaction at the SE|CAM interface, which is accompanied by the formation of polysulfide species. When using carbon additives, the polysulfide formation was significantly triggered and dominated the decomposition signals in S 2p spectra (integrated measurements of the whole cathode), not at least because these are also formed at the β-Li$_3$PS$_4$|carbon interface. These results agree very well with other observations on the oxidation of thiophosphate SEs[18,25]. Although it appears that part of this oxidation is reversible as a function of the cathode potential, and may even add up to the cathode capacity, there is also an irreversible part that reduces the practical long-term capacity of the cathode composite. Even gaseous SO$_2$ has been observed stemming from irreversible reactions between the oxygen from NCM622 and β-Li$_3$PS$_4$ at high potentials[26].

To the best of our knowledge, the kinetics of these degradation reactions has never been studied in detail. Jung et al.[27] reported that the chemical degradation at the Li$_6$PS$_5$Cl|NCM interface increases the impedance, but the degradation kinetics as a func-tion of state-of-charge (SOC) and the temperature has yet not been studied. This lack of information is on the one hand due to the fact that the analysis of the kinetics requires extended impedance studies of full cells; on the other hand, the thin degradation layer formed at the SE|CAM interface poses a great challenge for interfacial analytics.

In this paper, we report results for the degradation of the prototype LGPS|NCM622 interface through systematic electro-chemical impedance measurements as a function of time, SOC and temperature, complemented by time-of-flight secondary ion mass spectrometry (ToF-SIMS) measurements and analyses. We demonstrate that the interfacial degradation of the SE follows a typical parabolic rate law, and can be well described by a Wagner-type model for diffusion-controlled reactions. With increasing potential, the rate constant increases, as predicted by the model. Thus, the SOC defines the driving force (i.e., the difference in lithium chemical potential between NCM and LGPS), and con-trols the rate of impedance increase with time. Accelerated degradation in the high potential range (4.3–4.5 V vs. Li$^+$/Li), as also supported by ex situ ToF-SIMS electrode analyses, implies that oxidation reactions at these high voltages may give rise to higher rates of degradation. Clearly, the results indicate that the SOC not only dominates the driving force of the interfacial degradation but also influences the composition of the degrada-tion layer. The activation energy ($E_A$) of the degradation rate constant implies an additional diffusion process taking place at higher SOC.

## Results

**Electrochemical measurements**. All tests in this work were car-ried out with a homemade cell case (Supplementary Fig. 1). The NCM622 particle morphology and the cell configuration are shown in Supplementary Figs. 2–3. To study the degradation at the LGPS|NCM622 interface, cells were rested at open-circuit voltage ($V_{OC}$) for ~20 h after being charged up to a pre-defined voltage. Figure 1a illustrates the measurement procedure, for which impedance measurements were conducted every 30 min. According to the model applied by Sakuda et al. and Zhang et al.[22,28], the Nyquist plot was fitted with the equivalent circuit $(R(RQ)(RQ)(RQ)Q)$ shown in the inset in Fig. 1b. Three semi-circles in the high-frequency (~200 kHz), mid-frequency (~500 Hz), and low-frequency (~4 Hz) regions are mainly attributed to ionic transport in the composite cathode, the SE|CAM interface, and the anode (In/InLi) |SE interface, respec-tively. As shown in Supplementary Fig. 4, the In/InLi|SE interface appears to be stable. The three-electrode cells with an In/InLi reference electrode were built to separate the resistance con-tributions. Compared to the In/InLi|SE interfacial resistance, the SE|CAM interfacial resistance is clearly dominating the overall resistance (Supplementary Fig. 5), in particular, with low-cathode loading (Supplementary Fig. 6). Given the resistance increase, the deviation owing to the fitting process does not influence the major conclusions of this work (Supplementary Fig. 7).

The impedance plots in Fig. 1c clearly show that the resistance of the SE|CAM interface ($R_{CEI}$) increases significantly during resting. To evaluate the interfacial degradation quantitatively, a Wagner-type model for diffusion-controlled reactions was applied to describe the growth of the cathode degradation layer[19].

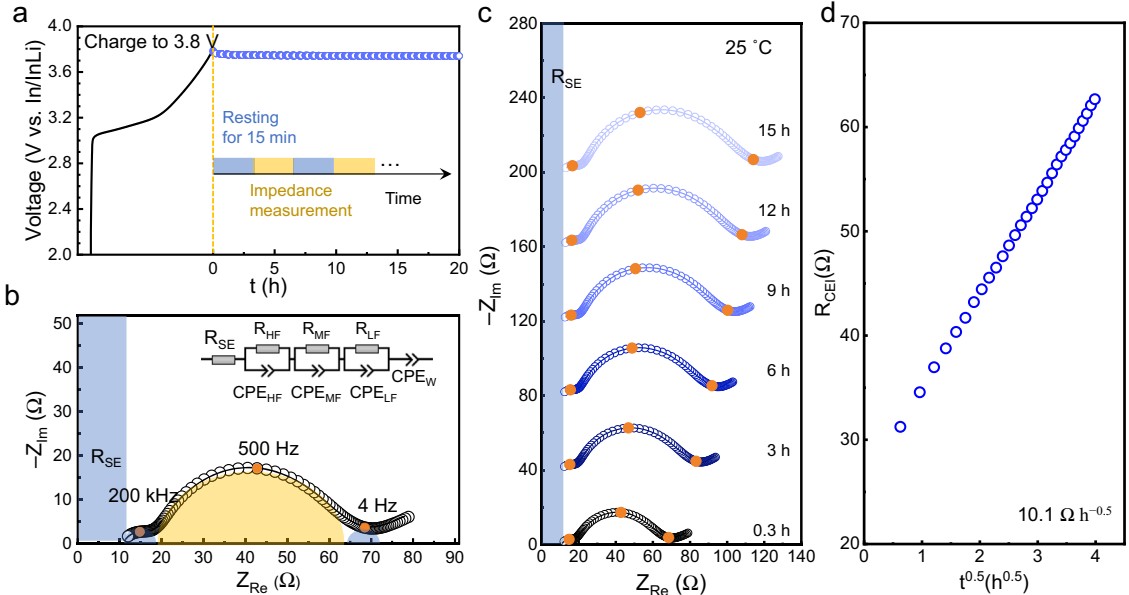

**Fig. 1 Impedance measurements of a full cell with LGPS|NCM622 composite cathode after charging up to 3.8 V vs. In/InLi (4.4 V vs. Li⁺/Li). a** The procedure of resting and impedance measurements. **b** Nyquist plot and the corresponding equivalent circuit used to evaluate the impedance data. **c** Nyquist plots of a typical cell with long-term resting. **d** The LGPS|NCM622 interface resistance $R_{CEI}$ as a function of the square root of time ($t^{0.5}$).

In this case, the model (Supplementary Fig. 8) leads to Eq. (1) with the assumption that charge is transported mainly by ions across the degradation layer (i.e., $\sigma_{Li^+} \gg \sigma_{e^-}$, details on the derivation of the equation are given in the Supporting Information).

$$R_{CEI} = \frac{1}{S\overline{\sigma_{CEI}}}\sqrt{\frac{V_m}{xF^2} \cdot \frac{\overline{\sigma_{Li^+} \cdot \sigma_{e^-}}}{\overline{\sigma_{Li^+} + \sigma_{e^-}}} \cdot \triangle\mu_{Li}} \cdot \sqrt{t} = \frac{1}{S\overline{\sigma_{CEI}}} \cdot k\sqrt{t} = k'\sqrt{t} \quad (1)$$

Here, $S$ denotes the contact area, $F$ represents Faraday's constant, $x$ denotes the number of moles of Li extracted from LGPS, and $t$ represents the resting time. The average ionic conductivity of the cathode|electrolyte interphase (CEI) layer is denoted as $\overline{\sigma_{CEI}}$, $V_m$ represents the average molar volume of the CEI. $\overline{\sigma_{e^-}}$ and $\overline{\sigma_{Li^+}}$ denote the mean partial electronic and ionic conductivity of the CEI layer. The driving force for the CEI growth, i.e., the difference of the chemical potential of lithium across the CEI, $\triangle\mu_{Li}$, is also included in the rate constant. The rate constants $k$ and $k'$ reflect the growth rate in terms of thickness and resistance, respectively.

We plotted the LGPS|NCM622 resistance against $t^{0.5}$ after collecting all $R_{CEI}$ values (Fig. 1d). $R_{CEI}$ increases linearly with $t^{0.5}$, as well described by Eq. (1). The growth of the degradation layer is driven by the lithium chemical potential difference between LGPS and NCM, i.e., across the degradation layer, at otherwise fixed thermodynamic conditions (i.e., temperature and electrode components). The lithium chemical potential in NCM is fixed by the molar fraction of Li, which corresponds to the SOC and follows electrochemical intercalation/de-intercalation.

Stepwise cyclic voltammetry (CV) measurements were performed to determine the stability window in full cells (InLi|LGPS| LGPS/NCM). As shown in Supplementary Fig. 9, the gap between positive and negative peaks reflects the polarization stemming from the internal resistance. The increasing overpotential demonstrates that the internal resistance grows remarkably with time and increasing cutoff voltage.

To investigate the impact of the SOC on interfacial degradation, impedance measurements were performed at different cutoff voltages. We define the SOC as 100% when the voltage reaches 3.9 V vs. In/InLi. In Supplementary Fig. 10, the voltage is

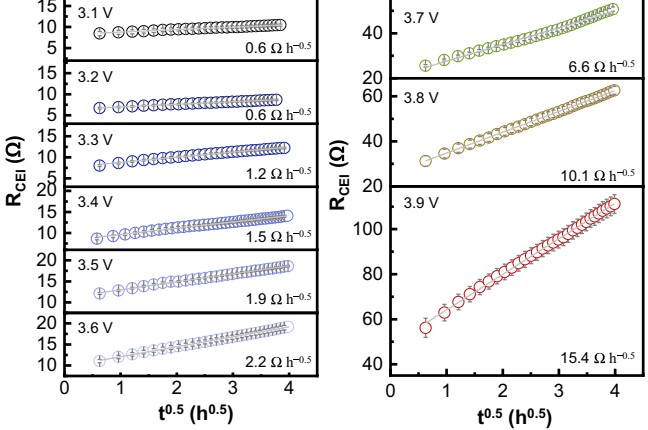

**Fig. 2 The voltage-dependence of interfacial resistance growth.** Linear fits of $R_{CEI}$ at the LGPS|NCM interface obtained from impedance data (see Supplementary Fig. 11) as a function of $t^{0.5}$ with different cutoff voltages (vs. In/InLi) at 25 °C. The error bars are determined by one standard deviation on the impedance fitting.

displayed as a function of specific capacity, SOC, and Li concentration in the cathode. Supplementary Fig. 11 presents impedance data with increasing resting time at the different cutoff potentials. The resistance of the LGPS|NCM interface increases (yellow regions in Supplementary Fig. 11) gradually with time, as shown for 3.8 V exemplary in Fig. 1c. As described by the model of diffusion-control reactions, $R_{CEI}$ increases linearly with $t^{0.5}$ at different SOC (Fig. 2) and does not saturate. Moreover, the interfacial resistance increases with elevating SOC, i.e., the degradation reaction is driven by the lithium chemical potential difference across the degradation layer, well in line with Eq. (1).

In order to quantitatively compare the growth rate at different SOC, we evaluated the rate constant $k'$ from the slopes of the $R_{CEI}$ fits. As shown in Fig. 3 and Supplementary Table 1, the slope increases weakly in the range of 3.1 to 3.6 V vs. In/InLi, suggesting that the LGPS|NCM interface degrades slowly in the lower potential range. In contrast, the slope increases to 6.6 Ω h⁻⁰·⁵ at

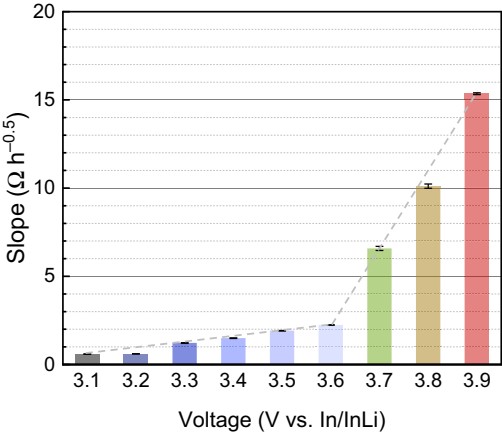

**Fig. 3 Quantitative comparison of the interphase resistance growth.**
Slopes of the linear fits of $R_{CEI}$ as function of the square root of time with different cutoff voltage at 25 °C. The error bars are determined by one standard deviation on the impedance fitting.

3.7 V, and increases to 10.1 and 15.4 $\Omega$ h$^{-0.5}$ at 3.8 and 3.9 V vs. In/InLi, respectively. Equation (1) predicts that the rate constant $k'$ should itself show a square root dependence ($k' \sim \triangle\mu_{Li}^{0.5}$) on the chemical potential difference across the degradation layer, i.e., roughly the cell voltage, if all other parameters remain constant. This appears to be the case for the cell voltages below 3.6 V vs. In/InLi. At higher potentials, the steeper increase of the rate constant indicates that the transport properties of the degradation layer also change with increasing SOC. As evident from Eq. (1), the electronic and ionic conductivity influence the growth rate as well, and the rate constant indicates that the composition of the degradation layer is also potential-dependent. These interconnected dependencies will be discussed in the following.

In addition to the increasing $R_{CEI}$, the capacitance of the corresponding $RQ$ element decreases simultaneously (Supplementary Fig. 12), which is well explained by the thickening of the resistive degradation layer. The capacitance drops quickly in the early period of interfacial degradation due to the parabolic relation between thickness and resting time. At the cutoff voltage of 3.1 V, the interfacial capacitance decays much faster with time. This trend indicates that the LGPS|NCM interface already degrades even at such a low SOC, however, yet not leading to a very thick CEI.

The galvanostatic intermittent titration technique (GITT) is most commonly used to investigate both thermodynamics and kinetics of electrode reactions. However, in the operation of SSBs, $V_{OC}$ is not exclusively dominated by the relaxation, i.e., the equilibration of the local lithium concentration, but is as well influenced by additional processes, such as interfacial degradation. Recently, our group reported that the combined effect of chemical diffusion and self-discharge could be simultaneously monitored with GITT (Eq. (2))[29]. In the following, we assume that the self-discharge is predominantly caused by interfacial reactions at the positive electrode, which lead to an oxidative decomposition of the SE and a reduction of the cathode.

$$V_{oc}(t) = V_{oc}^{Di}(t) + V_{oc}^{De}(V, t) \qquad (2)$$

Hereby, $V_{oc}(t)$ represents the measured open-circuit potential at time $t$, $V_{oc}^{Di}(t)$ and $V_{oc}^{De}(V, t)$ describe the thermodynamic open-circuit voltage (including relaxation by chemical diffusion (Eq. (3)) and voltage loss due to interfacial degradation (Eq. (4)). Semi-infinite relaxation was assumed for the diffusion kinetics under the condition that the diffusion layer is only partially extended throughout the electrode within the given

polarization time $t_{pol}$ (Eq. (3))[30]. The interfacial degradation is represented by the concentration-dependent thermodynamic factor $K_x$ and the lithium concentration, and the growth rate could be obtained with the introduction of the contact area and average molar volume of the degradation layer (Eq. (4)).

$$V_{oc}^{Di}(t) = V_0 - \frac{2}{\sqrt{\pi}} \frac{IK_xRT}{F^2Sc_0\sqrt{\widetilde{D}_{Li}^{app}}} \left[\sqrt{t} - \sqrt{t + t_{pol}}\right] \qquad (3)$$

$$V_{oc}^{De}(t) = \frac{-RTK_x}{xF}\triangle x = \frac{-RTK_x}{xF}\frac{S}{V_m}\triangle\xi = \frac{-RTK_x}{xF}\frac{S}{V_m}k\sqrt{t} \qquad (4)$$

With Eqs. (3) and (4), Eq. (2) can be written as:

$$V_{oc}(t) = V_0 - a\left[\sqrt{t} - \sqrt{t + t_{pol}}\right] + b\sqrt{t} \qquad (5)$$

Here, $V_0$ and $I$ denote the initial $V_{OC}$ and applied current, $R$ and $T$ denote the gas constant and temperature, $c_0$, $\widetilde{D}_{Li}^{app}$, and $t_{pol}$ represent the equilibrium concentration of lithium at given SOC, the apparent chemical diffusion coefficient of lithium and the duration of galvanostatic charge pulse, $\triangle\xi$ and $V_m$ represent the thickness and the average molar volume of the degradation layer, respectively.

The results obtained with GITT are depicted in Supplementary Fig. 13a. Galvanostatic charging (4 mA g$_{NCM}^{-1}$) and relaxation steps were applied, each step followed by impedance measurement. These steps were repeated until the upper cutoff potential reached 3.9 V (vs. In/InLi). Equation (5) was used to fit the relaxation curves (Supplementary Fig. 13b). Supplementary Fig. 14a displays the trend observed for the factors $a$ and $b$, with $a$ containing information on the chemical diffusion in the electrode and $b$ on the growth rate constant, respectively.

As shown in Supplementary Fig. 14b, the apparent chemical diffusion coefficient of lithium ($\widetilde{D}_{Li}^{app}$) determined from Eq. (3) is in the range of 10$^{-11}$ to 10$^{-13}$ cm$^2$ s$^{-1}$, which is in good agreement with the literature[29]. In the middle range of $V_{OC}$ (3.2–3.6 V), the higher $\widetilde{D}_{Li}^{app}$ is ascribed to high concentrations of both lithium and vacancies in the lithium-ion sublattice. The growth rate constant was semi-quantified with assuming the complete decomposition of LGPS and neglecting the changes in structure and molar volume (Supplementary Fig. 14c, details on the estimation are given in the Supporting Information). In the low-voltage range (3.0–3.2 V vs. In/InLi), the weak impact of chemical diffusion and interfacial degradation gives rise to relatively large fitting errors. The growth rate increases significantly with SOC, and reaches ~3.4 nm h$^{-0.5}$ at the highest open-circuit voltage of 3.9 V. We would like to note that the result calculated through electrochemical measurements is in very good agreement with previous experimental results[31]. Clearly, the semi-quantitative results further confirm that a higher SOC accelerates the degradation process.

With regard to the EIS data (Supplementary Fig. 15), it should be noted that the semi-circle in the intermediate frequency range is composed not only from the resistance of the CEI layer, but also from the charge transfer at the interface. In the beginning of the charging process (fully lithiated NCM), the large $R_{CEI}$ can be attributed to sluggish kinetics of the charge-transfer process. In the range from 3.1 to 3.5 V, the diffusion coefficient and the interfacial degradation remain almost constant, resulting in similar values for $R_{CEI}$. From 3.6 V on, $R_{CEI}$ increases due to the synergetic impact of the decreased $\widetilde{D}_{Li}^{app}$ and a thicker degradation layer.

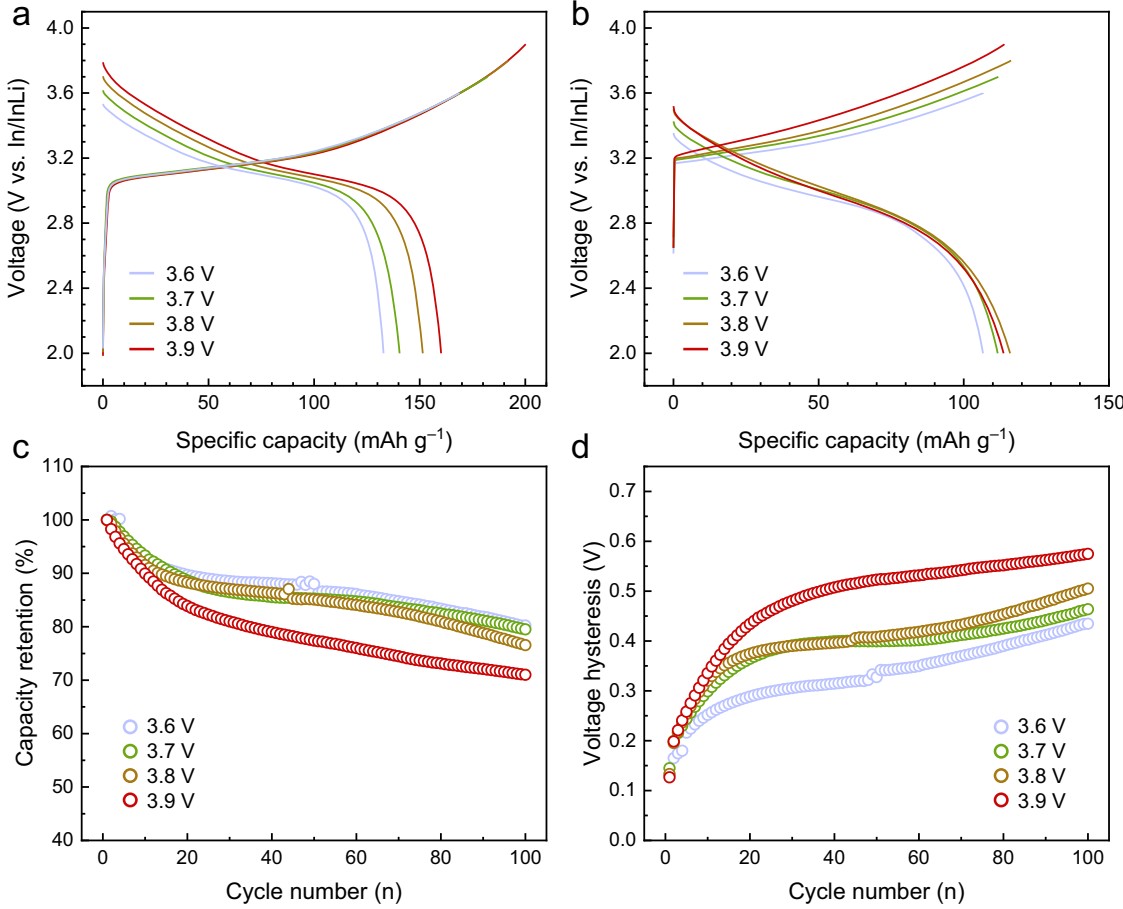

**Fig. 4 Galvanostatic charge–discharge tests of SSBs applying different upper cutoff voltage.** Charge–discharge curves of the **a** 1st and **b** 100th cycle. **c** Capacity retention and **d** voltage hysteresis of four cells in 100 cycles. The capacity retention in **c** is normalized with the corresponding discharge capacity of 1st cycle. The voltage hysteresis in **d** is calculated with the difference of average charge/discharge voltage.

**Long-term cycling performance**. The long-term cycling testing was also carried out with different upper cutoff voltages. Figure 4a displays the charge–discharge curves in the first cycle. The cells exhibit cutoff voltage-dependent charge capacities of 169 mAh g$^{-1}$ (3.6 V), 182 mAh g$^{-1}$ (3.7 V), 192 mAh g$^{-1}$ (3.8 V), and 200 mAh g$^{-1}$ (3.9 V), respectively. After discharging to the same voltage (2 V), the four cells deliver discharge capacities of 133 mAh g$^{-1}$, 140 mAh g$^{-1}$, 151 mAh g$^{-1}$, and 160 mAh g$^{-1}$, resulting in virtually similar Coulomb efficiencies of 78.7%, 77.1%, 78.9%, and 80.0%, respectively. The higher specific capacity implies that more lithium can be extracted/intercalated with a higher cutoff voltage. The galvanostatic charge/discharge curves after 100 cycles are displayed in Fig. 4b, the cells exhibit capacity fading due to the degrading interface. The remaining discharge capacities of the four cells are 107, 112, 116, and 114 mAh g$^{-1}$, respectively. To better compare the cycling stability, the capacity retention and the voltage hysteresis are evaluated. As is shown in Fig. 4c, the cycling stability is inversely correlated with the applied upper cutoff voltage, suggesting that a higher cutoff voltage leads to more severe degradation. In addition, the voltage hysteresis (Fig. 4d) and the $dQ/dV$ plots (Supplementary Fig. 16) of different cells demonstrate that strong degradation at high voltage further accelerates the growth of voltage polarization.

To assess the influence of other factors (i.e., chemo-mechanical volume change and irreversible phase transformation) on the cycling performance, the charge curves of the second cycle are shown in Supplementary Fig. 17. The capacities at 3.5 V reflect the fraction of the CAM in the composite cathode, ~4% capacity

loss was observed once the cell is charged up to 3.7 V. In consequence, the contact loss between SE and CAM particles, and irreversible phase transformation of CAM result in limited inactive regions[32]. In addition, microstructure images of the composite cathode at different states are displayed in Supplementary Fig. 18, which shows the absence of significant contact loss at the interface. By and large, regarding the impedance increase and capacity fading, the interface degradation by CEI formation is more pronounced than the chemo-mechanical effect in the first cycle.

**Chemical characterization**. ToF-SIMS was used to obtain complementary chemical insight into the degradation of the interface. In order to reduce local deviations in the chemical composition and secure more reliable results, we measured mass spectra in 10 different areas on each composite cathode surface and applied boxplots to compare the normalized signal intensities. It was already reported that a significant fraction of degradation products form close to the current collector (CC). Since the surface oriented towards the CC was investigated, the overlap of the decomposition processes at the CC|SE and the SE|CAM interface needs to be taken into account in the following when interpreting the experimental data[24]. In the context of ToF-SIMS studies on decomposition processes in NCM- and thiophosphate-based composite cathodes, the formation of polysulfide species and oxygenated phosphorous and sulfurous species has already been reported[24,31,33,34]. For this reason, we focused on signals which can be attributed to long-chain

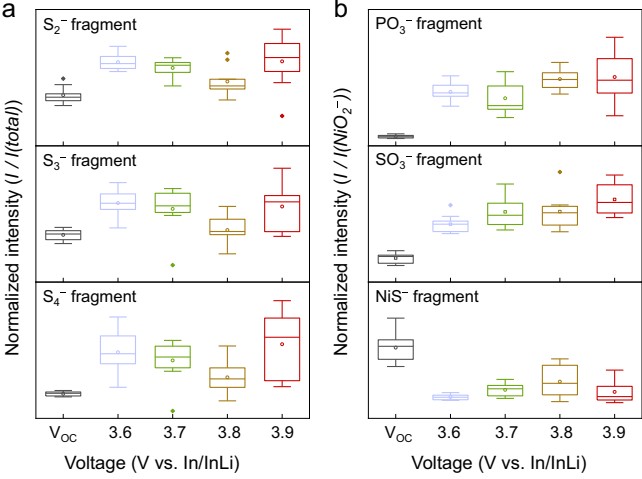

**Fig. 5 Boxplots of ToF-SIMS surface analysis.** Comparison of the normalized intensities of **a** $S_2^-$/$S_3^-$/$S_4^-$ and **b** $PO_3^-$/$SO_3^-$/$NiS^-$ fragments at the surface of cathodes with different SOCs. The intensities of $S_2^-$/$S_3^-$/$S_4^-$ signals were normalized with the total ion intensity, and the intensities of $PO_3^-$/$SO_3^-$/$NiS^-$ fragments were normalized in relation to the $NiO_2^-$ signal. The lines in boxes depict the median, lower and upper box limits indicate 25th and 75th percentiles, respectively. Whiskers extend to ±1.5 × IQR, and points are outliers.

polysulfide fragments ($S_x^-$, $2 \leq x \leq 4$) and oxygenated phosphorous and sulfurous species ($PO_3^-$ and $SO_3^-$) to gain insights into the SOC-dependent degradation reactions.

Figure 5a shows the results obtained for the polysulfide fragments. Compared with the uncycled electrode, the signal intensities of $S_x^-$ fragments increase in all four cases, indicating that the formation of long-chain polysulfides is part of the decomposition of LGPS. This is in accordance with reports for other thiophosphate-based SEs[18,24,25]. However, a SOC-dependence of the polysulfide formation is hard to determine. It seems that the amount of detected $S_x^-$ fragments decreases for 3.7 and 3.8 V and rises again for 3.9 V. In this context, the overlap of the decomposition processes with contributions from the CC| SE interfacial reaction must be considered. Therefore, the deviations between the post mortem samples could also be explained, for example, by deviations in the contact area between the CC and the SE and the associated different amounts of polysulfides formed. Consequently, the results for the polysulfide formation cannot be explicitly related to the SE|CAM interface here.

In contrast to $S_x^-$ fragments, an increasing amount of phosphate ($PO_3^-$) and sulfate/sulfite ($SO_3^-$) fragments can be referred to as the SE|CAM interface, since NCM is the only oxygen source within the composite cathode. To address possible differences in the fraction of NCM particles within the analysis area, we have chosen a normalization approach based on the specific NCM fragment $NiO_2^-$. In this way, detrimental effects due to different fractions of NCM particles within the analysis area can be minimized. Figure 5b shows the results obtained for phosphate and sulfate fragments. An increase of $PO_3^-$ and $SO_3^-$ fragments for the four post mortem samples confirms the interfacial degradation during charging. Moreover, the normalized intensities of $PO_3^-$ and $SO_3^-$ fragments further increase as the cutoff voltage is elevated, clearly revealing a SOC dependency for the formation of oxygenated phosphorous and sulfurous species.

At this point, we would like to note that carbon additives were not used, as is often the case. Therefore, it is basically possible that inactive CAM particles are present within the analysis areas, even

if the probability for an electrically conductive contact is high close to the CC. Nevertheless, to address this issue and to verify the results, ToF-SIMS investigations were performed again with composite cathodes containing 3 wt.% vapor-grown carbon nanofibers. In this way, enhanced utilization of the CAM is achieved. Supplementary Fig. 19 compares the results obtained for composite cathode with/without carbon additives.

For the samples with carbon fibers, higher intensities for polysulfide fragments can basically be detected after charging. This is in accordance with previous studies, which have demonstrated the reactivity of thiophosphate-based SEs with carbons, which can lead to the formation of polysulfides and other oxidized species[24,25,31,35]. Since the SOC-dependent trends in the fragment intensities with and without carbon additives are different, we assume that this is due to the detrimental overlap with the CC|SE interface. We speculate that (i) the amount of detected polysulfide fragments is dominated by this interface and (ii) the difference in the polysulfide fragment intensities can be explained by deviations in the contact area between the CC and the SE. Accordingly, we cannot draw conclusions on the polysulfide formation at the LGPS|NCM interface here. However, the total amount of polysulfide fragments (regardless of the location of their formation) does not appear to significantly increase by elevating the upper cutoff potential. This indicates that polysulfide formation occurs mainly at potentials below the chosen cutoff potentials. This is consistent with a report by Dewald et al.[18], who reported the practical oxidative stability limit of thiophosphate-based SEs to be in the range of 2.2 and 2.5 V (vs. In/InLi).

In contrast to $S_x^-$ fragments, the same trends can be observed for $PO_3^-$ and $SO_3^-$ fragments with and without the use of carbon additives. Accordingly, the normalized intensities of $PO_3^-$ and $SO_3^-$ fragments increase as the cutoff voltage is elevated, confirming the SOC dependency for the formation of oxygenated phosphorous and sulfurous species.

At this point, it should be noted that we have no evidence of the formation of transition metal sulfides, independent of the use of carbon additives. The signal that we have attributed to $NiS^-$ decreases in all cases compared to the uncycled electrode. This could indicate the consumption of transition metal sulfides during the oxidative reaction on charging, which is consistent with previous reports[24,31].

Overall, the results demonstrate two major degradation reactions in two voltage ranges: (i) the decomposition of LGPS and formation of polysulfide and other oxidized species (pyro-thiodiphosphates and meta-thiodiphosphates) solely by extraction of lithium in the low-voltage range (at all interfaces)—without oxygen from NCM; (ii) the interfacial reaction between NCM and LGPS involving oxygen in the high-voltage range.

**Interface degradation as a function of temperature.** According to the Wagner-type model, the chemical diffusion of lithium across the CEI layer determines the growth rate of the degradation layer. As the CEI is most probably primarily ion-conducting, electronic transport across the CEI layer is expected to be the rate-limiting step of interfacial degradation, suggesting the interfacial degradation shows exponential temperature dependence. Therefore, the $E_A$ in the Arrhenius-type notation of the parabolic rate constant $k$ (Eq. (6)) could reflect the threshold barrier of electron diffusion, i.e., chemical diffusion of lithium, at the interface.

$$k = Ae^{-\frac{E_A}{RT}} \tag{6}$$

To evaluate the temperature dependence, cells were rested at 10 °C and 40 °C after charging to different cutoff voltages at

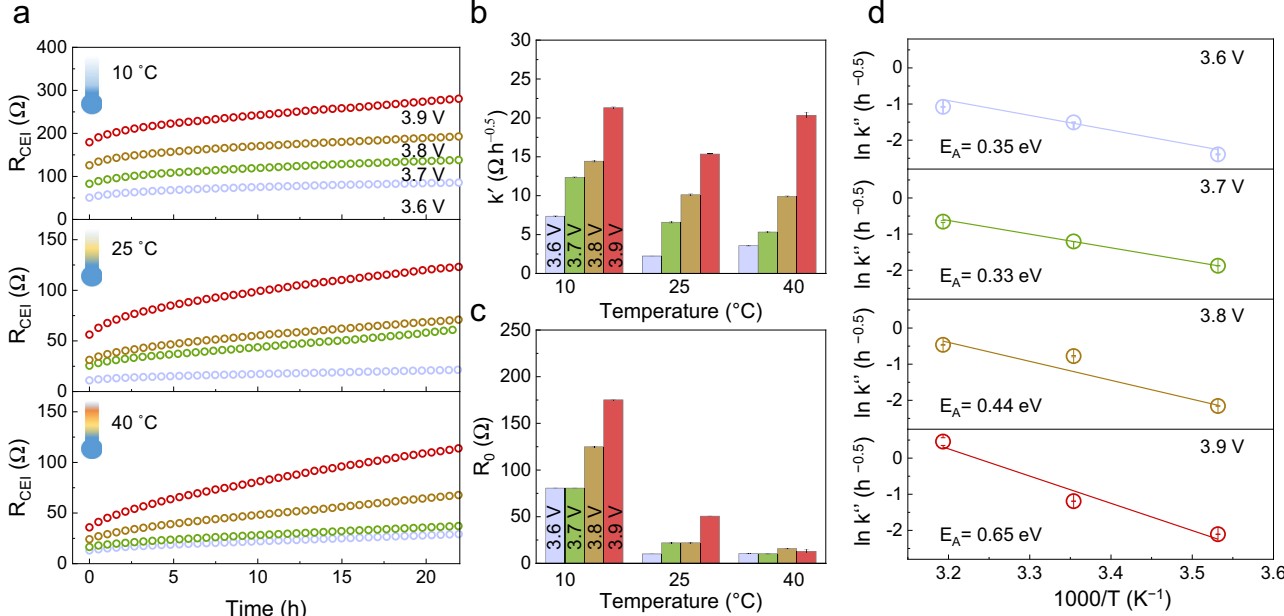

**Fig. 6 The temperature dependence of the interfacial resistance growth. a** The increase of interfacial resistance with time at different temperatures and SOC. **b** The slope ($k'$, rate constant) and **c** intercept ($R_0$, charge-transfer resistance) from the parabolic fitting of **a**. **d** The temperature dependence of normalized rate constant ($k''$). The activation energy was measured by the slope of the fitting line with Arrhenius equation. The error bars are determined by one standard deviation on the impedance fitting.

25 °C. The resistance evolution was monitored by impedance measurements. Figure 6a compares $R_{CEI}$ changing with time, SOC, and temperature. After fitting the $R_{CEI}$ change with the parabolic rate function, Fig. 6b, c display the rate constant ($k'$) and the charge-transfer resistance ($R_0$), respectively. According to Eq. (1), the rate constant, as determined from impedance data, is impacted by the temperature dependence of ionic conductivity as well. In order to compare the rate of the interfacial degradation at different temperatures, $k''$ was obtained by normalizing the rate constant $k'$ by the charge-transfer resistance $R_0$ (Eq. (7)).

$$k'' = \frac{k'}{R_0} = \frac{k}{\sigma_{CEI} S R_0} = \frac{k}{\xi_0} \qquad (7)$$

Where, $\xi_0$ denotes the thickness of the degradation layer at the beginning of the interfacial degradation. Here we assumed that the $R_{CEI}$ is $R_0$ before interfacial degradation takes place, and the temperature dependences of $R_{CEI}$ and $R_0$ are the same. The $E_A$ for the interfacial degradation was evaluated from the normalized rate constants as a function of temperature (Fig. 6d). At the cutoff voltages of 3.6 and 3.7 V, the activation energies are 0.35 and 0.33 eV, respectively. The slight decrease at 3.7 V could be attributed to the fitting error (Supplementary Table 2). The $E_A$ increases to 0.44 and 0.65 eV with elevating the cutoff voltages to 3.8 and 3.9 V, respectively. This reveals that additional oxygen diffusion processes at higher voltages may take place, leading to more severe degradation. This result again confirms that the applied voltage shows a strong influence on the degradation products and the growth kinetics.

## Discussion

The degradation of the LGPS|NCM interface is well explained by the Wagner-type model of diffusion-controlled interphase (CEI) growth. As a consequence, the degradation of LGPS is slowing down with time, but does not stop. Clearly, the increasing overpotential due to CEI formation leads to the capacity fading of SSB cells. The rate of the degradation reaction is SOC-dependent, highlighting the role of the lithium chemical potential difference

between NCM and LGPS as the driving force for the degradation reaction. Ideally, an ion-conducting and electron-insulating coating layer could reduce the driving force and suppress the degradation. Nevertheless, the electrochemical stability of most coated materials still suffers from imperfect coating. To avoid the interference of the inhomogeneous coating, we intentionally applied uncoated materials to obtain reference data. Also, the formed phosphate species at the interface have been used to mitigate the interfacial degradation[6,36]. Therefore, the in situ formed interphase could serve as a homogenous coating, which is of great interest for the future design of SSBs. With increasing temperature, the rate of degradation increases. As SSBs are often intuitively considered to operate even better at higher temperatures, this contradicts the naive expectation and will require care in introducing proper protection concepts. In fact, as the observed $E_A$ of degradation is higher than the $E_A$ of the transport processes involved in the charge and discharge steps, a temperature increase will always favor degradation relative to normal battery operation. Concerning the trade-off between degradation and transport processes, this implies the true advantage of SSBs might be low-temperature operation rather than high-temperature operation.

**Influence of SOC on the degradation layer**. Our ToF-SIMS results provide detailed insights into the decomposition processes within the composite cathode. Although the surface is partially inhomogeneous and the detrimental overlapping of different decomposition processes must be taken into account, the relative intensities of different fragments can be used to unveil the decomposition processes at the LGPS|NCM622 interface. In the following, we describe the degradation processes using the schematic diagram in Fig. 7. It must be noted, that this is a simplified picture of the complex interfacial reaction and we do not want to hypothesize on specific stoichiometries and compounds here but rather indicate probable interfacial processes.

Degradation reactions between NCM622 and LGPS take place already directly after physical contact, which is in agreement with

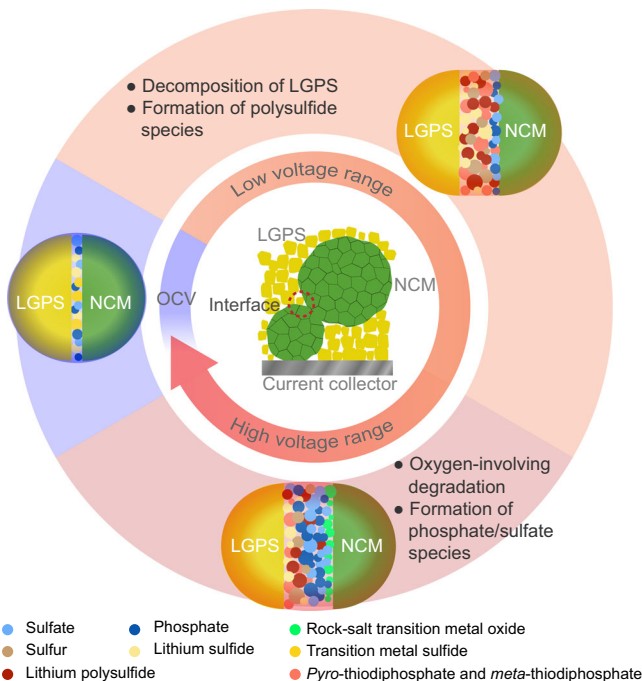

**Fig. 7 Schematic of the evolution of interfacial degradation at the LGPS| NCM622 interface.** In low-voltage range (3.1 V vs. In/InLi < E < 3.7 V vs. In/InLi), the electrochemical decomposition of SE takes place at the interface. In the high-voltage range ($E \geq 3.7$ V vs. In/InLi), the oxygen-involving degradation dominates the interfacial reaction.

theoretical calculations and experimental observations[24,27,31,37]. When charging the cell, interfacial degradation becomes more severe, as the driving force increases. At low potentials (3.1 vs. In/InLi < E < 3.7 V vs. In/InLi), "mild" oxidative degradation of LGPS takes place, i.e., by oxidation of sulfur species forming polysulfide and other oxidized species, as already described for other thiophosphate-based SEs[24,25,31,38]. At high potentials ($E \geq 3.7$ V vs. In/InLi), the interfacial reaction between the NCM and the SE becomes more pronounced and leads to the formation of oxygenated phosphorous and sulfur species (i.e., phosphates/phosphites, sulfates/sulfites) in the interfacial region (indicated by ToF-SIMS), which is in accordance to reports on other thiophosphate-based composite cathodes[24,31,33]. Our ToF-SIMS data show that the formation of oxygenated phosphorous and sulfurous species is SOC-dependent and is aggravated when the cutoff voltage is elevated. The oxygen released from the NCM lattice and the subsequent reaction with the SE can even lead to the formation of gaseous species such as $SO_2$ as described in literature[26]. In addition, the formation of a rock-salt-like oxide phase can be assumed, which further blocks the transport of lithium ions and electrons[39,40].

**Influence of SOC on the interfacial degradation kinetics.** The Wagner-type model describes the kinetics of interfacial degradation quite well. According to Eq. (1), the rate constant includes the driving force for the degradation reaction, which is the difference in lithium chemical potential across the degradation layer. The degradation products and their corresponding properties also depend on the SOC, which makes the SOC-dependence mathematically complicated. Although we cannot provide detailed and unequivocal information on the components within the CEI and their stoichiometric ratio, we performed a qualitative analysis as a basis to interpret the influence of SOC on the interfacial degradation kinetics. According to Eq. (1), apart from the difference in

lithium chemical potential, higher electronic conductivity and lower ionic conductivity trigger accelerated degradation. At low voltage (3.1 V vs. In/InLi < E < 3.7 V vs. In/InLi), the formation of polysulfide species decreases the ionic transport across the CEI layer, which results only in a slight increase of the growth rate with potential. Regarding the oxygen-involving degradation in the high-voltage range ($E \geq 3.7$ V vs. In/InLi), the highly resistive rock-salt region further blocks ionic transport through the interface, and induces the steep increase of the rate constant. In fact, the higher $E_A$ confirms a higher diffusion barrier due to the more resistive interfacial products.

**Influence of SOC on the electrochemical performance.** The electrochemical performance of SSBs is—neglecting at this point the anode and separator—the result of many factors occurring in the cathode composite. At least three key factors affecting the electrochemical performance: structural changes of the CAM[32], chemo-mechanical effects[29,41], and interfacial degradation[24,31] due to oxidation of the SE. In the present work, we identify oxidation of the SE as the most critical factor in increasing the overpotential and deteriorating the electrochemical performance, once the SE is not properly protected by a coating of the CAM.

Although it has been known that thiophosphate electrolytes are oxidized by cathode materials and form resistive interlayers, our quantitative work provides important insight into the kinetics of interfacial degradation. Using EIS and other electrochemical measurements, we demonstrate that LGPS as prototype (and model-type) SE continuously degrades as a function of time, following a classical $t^{0.5}$-dependence of diffusion-controlled reaction. While the absolute rate depends on the SOC of the SSB, degradation of unprotected SE never stops and will even increase during operation at elevated temperatures. This result highlights the need for a proper protective coating of NCM materials, not only avoiding the direct contact and reaction with the SE, but also avoiding electronic charge transfer in general. Obviously, the CEI components are not electronically insulating enough, so the chemical diffusion of lithium across the CEI facilitates continuous degradation. The quantitative results may help to better judge the impact of a protective coating, as this should lead to a major reduction of the parabolic rate constant of CEI formation.

The interfacial degradation of the LGPS|NCM composites in SSB is assessed quantitatively. Impedance and GITT measurements were applied to monitor the growth process of the degradation layer, which fits well with a diffusion-controlled model. ToF-SIMS results unveil that two different types of degradation reactions take place in two potential ranges at the LGPS|NCM interface. By and large, interfacial degradation is a complicated process influenced by time, SOC, and temperature. Regarding the measurement conditions, SOC is a reasonable measure of the driving force for the growth of the degradation layer. It should be noted that the LGPS|NCM interface is not stable even upon physical contact. In the low cutoff voltage range (3.1 V vs. In/InLi < E < 3.7 V vs. In/InLi), the extent of interfacial degradation increases gradually, but remains insignificant as the SE is only degraded by lithium extraction. With further elevating the SOC, the high voltage ($E \geq 3.7$ V vs. In/InLi) triggers oxygen-involving degradation and the formation of rock-salt phase, the latter of which further accelerates the growth of the interfacial resistance. Our model work elucidates the critical role of interfacial stability and rational coating design in SSBs.

## Methods
**Synthesis.** LGPS was synthesized via a solid-state method[22]. $Li_2S$ (Simga-Aldrich), $P_2S_5$ (Simga-Aldrich), GeS (Simga-Aldrich), and S powder (Simga-Aldrich) were mixed with the stoichiometric ratio. 5% excess S was also added into the precursors

to compensate S loss during synthesis. The mixture was fully blended with $ZrO_2$ milling balls in a planetary ball milling machine. The rotation speed was 500 rpm, and the rotation time was set as 40 h (15 min cooling time was followed by 10 min ball milling to prevent over temperature). After ball milling, the mixture was sealed into ampules (~1 g per ampule) in an argon-filled glovebox, and then the ampules were evacuated and sealed. Subsequently, the sealed ampule was annealed at 500 °C for 30 h with a heating rate of 28 °C h$^{-1}$. The as-obtained LGPS was stored in a glovebox with $H_2O$ and $O_2$ contents below 0.1 ppm.

**Cell assembly**. The cells were built in an Ar-filled glovebox with $H_2O$ and $O_2$ contents below 0.1 ppm. In all, 60 mg LGPS was put into the PEEK (Polyether ether ketone) cylinder of a homemade cell (Supplementary Fig. 1), and pressed into a pellet with the hand press. The $LiNi_{0.6}Mn_{0.2}Co_{0.2}O_2$ (NCM622, BASF) particles are polycrystalline (Supplementary Fig. 2), and their size varies in the range of 2–8 μm. The composite cathode was prepared by blending LGPS and NMC622 powders with a mass ratio of 3:7, then the mixture was hand-grinded in an agate mortar for 20 min. 12 mg composite cathode powder was then added to one side of the LGPS pellet. The thickness of the composite cathode is ca. In all, 65 μm (Supplementary Fig. 3). For ToF-SIMS characterization, 3% extra carbon fibers were added into the composite cathode. Afterwards, the cell was uniaxially pressed with higher pressure of three tons for 3 min (~380 MPa). Thin In foil (Alfa Aesar, 99.99%, 8 mm diameter, ~100 μm thickness) and Li foil (Albemarle, Rockwood Lithium GmbH, 99.9%, 4 mm diameter, ~120 μm thickness) were added at the other side to form the In/InLi anode. In order to equilibrate the In/InLi anode, the assembled cell was fixed in a homemade aluminum frame with ~28 MPa for 1 h. During cycling, the cell was fixed in the framework with higher pressure (~70 MPa).

The three-electrode cell was built with the same cell case and a 2-part PEEK cylinder. To keep the separator intact during processing, 80 mg LGPS was put into the PEEK cylinder, and pressed into a pellet with the hand press. A 2 mg In foil rolled on a thin stainless steel wire was put on the surface of LGPS pellet to serve as the reference electrode. Another 80 mg LGPS was added on the top of the reference electrode, and pressed into an integral separator (10 mm diameter, ~1.4 mm thickness). The anode and cathode in 3-electrode cells are same as that in two-electrode cells.

To check the stability of LGPS against the In/InLi electrode, we built a Li|LGPS| In cell and monitored the interfacial resistance during lithiation. An In foil (0.1 mm in thickness, 6 mm in diameter) was applied as a working electrode, and excess Li was added at the anode side to ensure complete lithiation. A low pressure (~28 MPa) was applied to avoid short-circuit.

**Electrochemical tests**. To evaluate the cycling stability, stepwise CV was conducted in the voltage range from 2.0 to 3.9 V (vs. In/InLi). The CV measurement was first carried out in the voltage range from 2.0 to 3.0 V for two cycles, followed by increasing the cutoff voltage from 3.0 to 3.9 V with a stepwise of 0.1 V.

EIS measurements were conducted to study the interfacial evolution. The measurements were carried out in a 25 °C climate chamber (Binder) to prevent the changes in impedance due to the temperature effect. The cell was charged to a certain voltage with a current density of 0.21 mA cm$^{-2}$ (0.1 C), and resting for 24 h. During resting, we carried out EIS once every ~30 min. The impedance spectra were measured by applying 10 mV amplitude superimposed to the open-circuit voltage in a frequency range of 1 MHz–0.1 Hz with 15 points per decade. All Nyquist plots were fitted with $R(RQ)(RQ)(RQ)Q$ equivalent circuit. According to the fitting results, we collected the resistance increase of the cathode/electrolyte interface, and plotted the resistance with the square root of time. The variable-temperature tests were performed in a climate chamber (Weiss Klimatechnik) with different temperatures to evaluate the kinetics and $E_A$ of the interfacial degradation. After charging to the upper cutoff potentials at 25 °C, the cells rest at 10 and 40 °C. The interfacial resistance was evaluated at different temperatures and SOC.

The GITT was applied to evaluate the side reactions during cycling. We first charged the cell for 0.5 h, followed by resting for 1 h and conducting the impedance measurement, and repeated the process until the voltage is over 3.9 V vs. In/InLi. Subsequently, we discharged the cell in a similar procedure until the voltage is below 2 V vs. In/InLi. Galvanostatic charge/discharge measurements were carried out to evaluate the cycling stability. The cells were charged to different upper cutoff potential (3.6, 3.7, 3.8, and 3.9 V vs. In/InLi) with a current density of 0.21 mA cm$^{-2}$ (0.1 C), followed by discharging to 2 V. In order to quantitatively compare the capacity fading, four cells were normalized with the capacity at 3.5 V.

The electrochemical tests were conducted with the VMP300 electrochemical workstation by Bio-Logic Science Instrument SAS.

**Characterizations**

*ToF-SIMS*. To avoid exposure to the ambient environment, the sample transfer process from the glovebox to the SIMS instrument was conducted with a Leica EM VCT500 shuttle (Leica Microsystems GmbH) under the exclusion of air. ToF-SIMS characterizations were carried out with a Hybrid SIMS M6 system (IONTOF GmbH, Münster, Germany). All measurements were conducted in negative ion mode, and $Bi_3^+$ ions (60 keV) were applied as primary ion species. To compare the difference in the mass spectra, the spectrometry mode (bunched mode) of the Bi-gun with a beam aperture of 1,100 μm was chosen to analyze the surface of pellets, which enables a high

mass resolution (FWHM $m/\Delta m > 6000$ @ $m/z = 31.97$ (S$^-$)). For charge compensation, the sample was flooded with low energetic electrons and argon at the main chamber pressure of $5 \times 10^{-6}$ mbar. The analysis area was set to $150 \times 150$ μm$^2$ and was rasterized with $256 \times 256$ pixels. The cycle time was 100 μs. The measurements were stopped at a primary ion dose of $10^{12}$ ions/cm$^2$. In all, 10 mass spectra were collected in different regions on the surface of each pellet to make the results reproducible and reliable. For the measurement and data evaluation, the Surface Lab 7.1 software (IONTOF GmbH, Münster, Germany) was used.

*SEM*. Top-view images of pristine and cycled pellets were obtained by a Merlin high-resolution scanning electron microscope (Carl Zeiss AG, Germany) at an acceleration voltage of 5 kV. Energy-dispersive X-ray (EDX) spectroscopy of the cross-sectional view of the cell was carried out using an XMAX EXTREME EDX detector (Oxford Instruments, U.K.) at an acceleration voltage of 7 kV. The samples were transferred with a Leica EM VCT500 shuttle (Leica Microsystems GmbH).

## Data availability

All data generated that support the findings of this study are included in the manuscript and supplementary information file.

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

## Acknowledgements

The authors acknowledge the financial support by the joint project InCa (project 03XP0228C) within the German-Japanese cooperation program of BMBF (Federal Ministry for Education and Research) and NEDO (New Energy and Industrial Technology Development Organization). Furthermore, the authors thank Jonas Hertle for the valuable discussion on the three-electrode cell results.

## Author contributions

T.T.Z. and J.J. conceived the idea. D.S. and J.J. supervised all aspects of the research. T.T.Z. synthesized LGPS and carried out the electrochemical tests with help from R.R., R.J.P., and S.H. F.W. and M.R. performed the ToF-SIMS measurements and conducted data analysis. T.T.Z. collected the SEM images. T.T.Z., R.K., and J.J. discussed the mechanism of interfacial degradation. T.T.Z. wrote the first draft of the manuscript. All other authors revised and approved the manuscript.

## Funding

## Competing interests

The authors declare no competing interests.
