## [Peer Review File · Nature Communications]

REVIEWER COMMENTS

Reviewer #1 (Remarks to the Author):

In this work, the authors analyse and quantify the degradation process occurring at the SE|CAM interface in the LGPS|NCM cathode composite, confirming a Wagner-type diffusion-controlled model. The work depicted in this manuscript presents novelty and relevance in the field as well as good overall execution. This reviewer thus suggests publication of this work in Nature Communications after addressing the following comments aimed at improving the quality and robustness of the work.

The authors should include information on the diameter and thickness of the employed cathode composite electrode.

In Figure S4 the authors show the Nyquist plots for the time-evolution of the EIS measurements at different resting potentials. At low voltages the three frequency regions that the authors describe can be clearly seen as separate semicircles, however, at higher voltages (~ 3.5 V and above), the low (SE|anode) and medium frequency (CEI) regions seem to overlap. The authors should confirm that their fitting model can unambiguously separate both contributions if these present different time domains, and do not correlate within the fitting process.

The voltage hysteresis is informative regarding the overall increases of the interfacial resistance, however, I recommend the authors to look at the dQ/dV plots where they could distinguish the voltage hysteresis for each individual redox event/phase transformation occurring at the NCM CAM, and particularly interesting for the higher voltage redox peak where the oxygen transfer from the NCM material towards the interphase formation could be taking place.

The addition of standard deviations in some of the data is appreciated, however, the authors should homogenise this to other parts of the manuscript where calculated parameters are missing their standard deviations.

The authors mention the oxidation of the sulphide electrolyte involving oxygen transfer from the NCM material. In this regard, ex-situ diffraction data (and/or high-resolution microscopy) could be interesting in order to analyse any possible Ni diffusion to Li layers, as it has been observed during oxygen release, as well as NCM phase transformations together with possible rock salt or spinel phases formation. These measurements could also inform on the interphase formed if this presents any long-range crystallographic order. Additionally, and to further confirm and understand the nature of the decomposition products forming the interphase, the authors could use vibrational spectroscopy to shed light on this aspect.

The In/In-Li electrode has been used in the literature with sulphides electrolytes as this possess lower (if any) reactivity compared with Li metal. However, since the formation of the alloy is still not yet standardised, with different Li/In ratios impacting the electrode performance (see <https://doi.org/10.1002/batt.201800149>), it would be appropriate if the authors could include a CV of their SE|In/In-Li setup to confirm the absence of any reactivity and the stability of the interface when swapping the cell to higher voltages to confirm that the electrode has low polarisation and does not contribute to the authors' analyses of the SE|CAM interface.

The authors recommend the use of electronically insulating coatings for the NCM material, this would, however, after the power capabilities of the cathode electrode. Do the authors think that a coating of the sulphide particles could be an alternative pathway to decrease the degradation at the CAM|SE interface instead?

To conclude, the authors should comment on and confirm the reproducibility of their study. The

authors mention the use of four replicas for the capacity fading test to 3.5 V, was the dispersion of the capacity in these cells low enough to expect the same in the other cell and electrochemical tests?

Dr Marco Amores, Austrian Institute of Technology

Reviewer #3 (Remarks to the Author):

The work aims to understand the interface reaction between NCM622 and LGPS vs SOC and temperature. Systematic electrochemical impedance and ToF_SIMS measurements were used to understand the kinetics and chemistry of the interface reaction. However, I can only know how to evaluate the analysis and conclusion of the work after the following concerns are addressed.

1) My main concern is whether the kinetics disclosed by Fig. 1, 2, 3 are dominated by the interface degradation of LGPS|NCM, as suggested by the authors. It seems to me some control experiments are missing here:

a) What about the voltage degradation of LGPS itself? If NCM is removed, with the cathode made only by LGPS and and some conductive medium, such as VGCF used by the authors, how will the corresponding Fig. 3 look like?

b) What about the influence of the anode interface between In/InLi and LGPS? During OCV at 3.x V will the anode interface change with time and thus change the impedance?

In fact in the charged state, since there is already Li migrated to the anode side, the anode interface stability really depends on the status of the migrated Li, e.g., if it is all alloyed with In or some of the Li is plated as Li metal without being well alloyed. As mentioned also by the authors, LGPS is known to be unstable with Li metal at least, so it is also likely that the kinetics discussed here mainly unveils the anode interface behavior instead.

I thus would suggest a control experiment. Use Li₄Ti₅O₁₂ mixed with LGPS as the anode, and make the anode capacity excessive wrt cathode NCM capacity, so intercalation at the anode is secured. Will the authors still see the same kinetics as described in Fig. 1-3? LTO mixed with LGPS was actually used in two of the authors' previous Nat. Energy paper ("normal type" in Table S4 of that paper), so I believe this experiment is doable by the team.

2)It's not mentioned in the Method what type of NCM622 was used, single crystal or polycrystalline? Tap density or particle size? Commercial product or synthesized?

3)It's not clear from the description in Method about the anode assembly. Is that LGPS|In(100um)|Li(120um)?

4)It's mentioned 12 mg composite cathode powder, but what is the cathode active material loading (mg/cm²)? From the 3:7 LGPS:622 and the anode In diameter of 8mm, it seems the authors are using a high loading of at least 16 mg/cm². This means a lot of Li will be migrated to the anode side in charge, and insufficient alloying with In together with Li metal plating might be very possible, so that the anode interface stability could be an issue.

What about if the authors lower the cathode loading significantly, such as by 5 to 10 times? Will the same kinetics in Fig. 1-3 still be observed?

Author's Response to Reviewers:

Reviewer 1

In this work, the authors analyse and quantify the degradation process occurring at the SE|CAM interface in the LGPS|NCM cathode composite, confirming a Wagner-type diffusion-controlled model. The work depicted in this manuscript presents novelty and relevance in the field as well as good overall execution. This reviewer thus suggests publication of this work in Nature Communications after addressing the following comments aimed at improving the quality and robustness of the work.

Response: We thank the reviewer for the strong endorsement of our work. Please find the point-by-point responses to the comments below.

Reviewer: The authors should include information on the diameter and thickness of the employed cathode composite electrode.

Response: Thank you for this suggestion. The diameter and thickness of the cathode composite are 10 mm and ca. 65 μm , respectively.

Changes in the manuscript: We added a cross-sectional SEM image in the Supplementary Information. As shown in the cross-sectional SEM image (Supplementary Figure 3a) and the corresponding element mapping (Supplementary Figure 3b), the structure of the SSB is well visible.

Supplementary Figure 3. (a) Cross-sectional SEM image and (b) the corresponding element mapping of the full cell.

Reviewer: In Figure S4 the authors show the Nyquist plots for the time-evolution of the EIS measurements at different resting potentials. At low voltages the three frequency regions that the authors describe can be clearly seen as separate semicircles, however, at higher voltages (~ 3.5 V and above), the low (SE|anode) and medium frequency (CEI) regions seem to overlap. The authors should confirm that their fitting model can unambiguously separate both contributions if these present different time domains, and do not correlate within the fitting process.

Response: We agree that the overlapping regions pose a challenge to distinguish the contributions from (anode) SEI and (cathode) CEI layers. We therefore conducted additional experiments with 3-electrode cells including a miniaturized reference electrode (developed in our lab in an independent study). The fitting results of WE vs. CE and WE vs. RE spectra match well. This result is the best possible confirmation for our analyses based on 2-electrode cells in the present manuscript.

Changes in the manuscript: We built the 3-electrode cell to differentiate the resistance contribution from anode|SE and SE|cathode interfaces, which rarely has been reported so far for SSB (with a few exceptions). A tiny piece of In foil (ca. 2 mg) was embedded in the separator to serve as "precursor" for the reference

electrode. Before cycling, the In electrode was lithiated with a weak current (30 μA) to form $\text{Li}_{0.1}\text{In}$, essentially being In/(InLi) two-phase mixture with a fixed reference potential). Similar to the 2-electrode cell, the 3-electrode cell was charged to 3.7 V and 3.9 V vs. In/InLi. During resting, the WE vs. CE, WE vs. RE and CE vs. RE impedance spectra were collected (Supplementary Figure 5).

With aid of such 3-electrode cell, the interfacial resistances of the In/InLi|LGPS anode and LGPS|NCM622 cathode interfaces can be well differentiated. Compared to the anode resistance (*i.e.*, CE vs. RE impedance), the cathode resistance (*i.e.*, WE vs. RE impedance) is clearly dominating the cell resistance (*i.e.*, WE vs. CE impedance). Furthermore, the resistance change of the cathode interface is more pronounced. This phenomenon fits well in the cases of 3.7 V (Supplementary Figure 5a-c) and 3.9 V (Supplementary Figure 5d-f) resting experiments.

As shown in the Nyquist plots, the 3-electrode cells show generally a higher resistance. This can be attributed to the interference of RE, due to its geometry, and the thicker separator that has to accommodate the RE (J. Electrochem. Soc. 2019, 166, A1550-A1557; J. Electrochem. Soc. 2002, 149, E166.).

As the reviewer suggested, the fitting results of WE vs. RE and WE vs. CE impedance spectra were compared in Supplementary Figure 6. The fitted R_{CEI} results fits well for 3.7 V cell, while the deviation in R_{CEI} is larger in the case of 3.9 V cell. We attribute this to the growing overlap at low frequencies. However, this only concerns the measurement at the maximum cutoff voltage. Compared to the trend in resistance increase, this deviation does not influence the major conclusions of our work, neither qualitatively nor quantitatively.

Supplementary Figure 5. EIS measurements of the 3-electrode cell during resting at 25 °C. The cell was charged to 3.7 V (a-c) and 3.9 V vs. In/InLi (d-f). (a and d), (b and e) as well as (c and f) show the impedance spectra and their corresponding fitting spectra of WE vs. CE, WE vs. RE and CE vs. RE, respectively.

Supplementary Figure 6. Comparison of the fitting results of R_{CEI} based on the WE vs. CE and WE vs. RE impedance spectra. (a-b) show the fitting results of cells after charging to 3.7 V and 3.9 V vs. In/InLi, respectively. The error bars were marked to show the deviation due to the fitting process.

Reviewer: The voltage hysteresis is informative regarding the overall increases of the interfacial resistance, however, I recommend the authors to look at the dQ/dV plots where they could distinguish the voltage hysteresis for each individual redox event/phase transformation occurring at the NCM CAM, and particularly interesting for the higher voltage redox peak where the oxygen transfer from the NCM material towards the interphase formation could be taking place.

Response: We thank the reviewer for this suggestion, and we fully agree that the dQ/dV plots are useful to determine the voltage polarization and interfacial resistance. One also should note that the SOC (*i.e.*, upper cutoff voltage in our experiments) influences the interfacial resistance as well. In order to compare the polarization effects, we chose the charging (delithiation) steps to plot dQ/dV curves. We plotted the dQ/dV curves of the 1st, 20th, 60th and 100th cycles (Supplementary Figure 16). The oxidative peaks shift to higher voltage during cycling, which indicates that the voltage polarization increases gradually. In addition, a higher cutoff voltage also leads to a stronger polarization. Considering the same SOC before charging (full-lithiated state), the voltage polarization in Supplementary Figure 16 is mainly dominated by the CEI resistance. This result supports the calculated results based on average charge/discharge voltage.

Changes in the manuscript: We added the following graph to the SI and added the following sentence to the main text: "In addition, the voltage hysteresis (Supplementary Figure 15d) and the dQ/dV plots (Supplementary Figure 16) of different cells demonstrate that strong degradation at high voltage further accelerates the growth of voltage polarization."

Supplementary Figure 16. The dQ/dV curves of four cells with different cutoff voltages (vs. In/InLi, light blue: 2 V - 3.6 V, green: 2 V - 3.7 V, yellow: 2 V - 3.8 V, red: 2 V - 3.9 V).

Reviewer: The addition of standard deviations in some of the data is appreciated, however, the authors should homogenize this to other parts of the manuscript where calculated parameters are missing their standard deviations.

Response: Thank you for the comment.

Changes in the manuscript: We have added the standard deviation in Figure 2.

Fig. 2 Linear fits of R_{CEI} at the LGPS|NCM interface obtained from impedance data (see Supplementary Figure 3) as function of $t^{0.5}$ with different upper cutoff voltages (vs. In/InLi) at 25 °C (standard deviation is shown in grey).

Reviewer: The authors mention the oxidation of the sulphide electrolyte involving oxygen transfer from the NCM material. In this regard, ex-situ diffraction data (and/or high-resolution microscopy) could be interesting in other to analyse any possible Ni diffusion to Li layers, as it has been observed during oxygen release, as well as NCM phase transformations together with possible rock salt or spinel phases formation.

These measurements could also inform on the interphase formed if this presents any long-range crystallographic order. Additionally, and to further confirm and understand the nature of the decomposition products forming the interphase, the authors could use vibrational spectroscopy to shed light on this aspect.

Response: Thank you for the comment. Oxygen release and phase transformation of NCM have been widely investigated in liquid-based LIBs, as well as in SSBs. (Nat. Comm. 2014, 5, 3529; Adv. Energy Mater. 2014, 4, 1300787; ACS Appl. Mater. Interfaces 2020, 12, 20462). Based on these references, we believe that the mechanism of phase transformation in the high voltage range (> 4.3 V vs. Li^+/Li) has been well unveiled. In our case, we focus on the kinetics of the transient process of LGPS|NCM interface degradation and in particular on its SOC-dependence, aiming for a quantitative analysis. The detailed chemical analysis of the decomposition products is beyond the scope of this paper, but has been treated in a number of recent papers from our group (papers by R. Koerver et al., e. g. J. Mater. Chem. A 5 (2017) 22750; and F. Walther et al., e. g. Chem. Mater. 32 (2020) 6123)

Reviewer: The In/In-Li electrode has been used in the literature with sulphides electrolytes as this possess lower (if any) reactivity compared with Li metal. However, since the formation of the alloy is still not yet standardised, with different Li/In ratios impacting the electrode performance (see <https://doi.org/10.1002/batt.201800149>), it would be appropriate if the authors could include a CV of their SE|In/In-Li setup to confirm the absence of any reactivity and the stability of the interface when swapping the cell to higher voltages to confirm that the electrode has low polarisation and does not contribute to the authors' analyses of the SE|CAM interface.

Response: Thank you for the suggestion. In comparison with the Li metal anode, the In/In-Li electrode shows indeed a better stability against sulfide-based SE (Batteries & Supercaps 2019, 2, 524.). However, we agree with the reviewer that issues related to the kinetics of the In/InLi anode are often underrated.

Changes in the manuscript: To check the stability of LGPS against the In/InLi electrode, we built a Li|LGPS|In cell and monitored the interfacial resistance during lithiation. An In foil (0.1 mm in thickness, 6 mm in diameter) was applied as working electrode, and excess Li was added at the anode side to ensure complete lithiation. We performed a constant current mode ($I = 50 \mu\text{A}$) to lithiate the In foil. As shown in the **Supplementary Figure 4a**, the first voltage plateau corresponds to the formation of the InLi phase. The duration of the first voltage plateau is 95.6 h, which corresponds to a capacity of ca. 4.78 mAh. The amount of In (n_{In}) can be calculated as:

$$n_{\text{In}} = \frac{V}{V_{\text{m}}} = \frac{0.1 \times \pi \times 3^2 \text{ mm}^3}{15.71 \text{ cm}^3 \text{ mol}^{-1}} = 180 \mu\text{mol}$$

V denotes the volume of the In foil. V_{m} represents the molar volume of In. The theoretical lithiation capacity (InLi phase) can be calculated as:

$$Q = n_{\text{In}} F = 180 \times 10^{-6} \text{ mol} \times 96488 \text{ C/mol} = 17.3 \text{ C} = 4.82 \text{ mAh}$$

As shown in **Supplementary Figure 4a**, the theoretical result coincides with the experiment result. At the beginning of lithiation, the potential is 579 mV. After alloying for 90 h, the potential decreases to 558 mV. According to the literature, the thermodynamic voltage plateau is 620 mV. We checked the impedance results at different points. As shown in **Supplementary Figure 4b**, the semi-circle at low frequency (ca. 4 Hz) corresponds to the In/InLi|LGPS interfacial resistance. The resistance remains stable during the long-term lithiation process, even after the formation of the Li_3In_2 phase. To prevent a short-circuit with the Li metal anode, we applied a relatively low pressure of 28 MPa. The low pressure gives rise to a high resistance of ca. 120 Ω . In contrary to the In/InLi|LGPS interface, the Li|LGPS interfacial resistance at higher frequency shows a significant increase during discharging. The high reactivity of Li leads to continuous SEI growth. In addition, Li vacancies and pores form at the interface of the Li anode, and the decreasing contact area plus the SEI give rise to a high interfacial resistance.

In addition, the 3-electrode EIS measurements in Supplementary Figure 5c and f also confirm the stability of the In/InLi|LGPS interface.

Supplementary Figure 4. (a) The voltage-time profile and (b) Nyquist plots of the Li|LGPS|In cell during lithiation process. The characteristic frequency (ca. 4Hz) of the In/InLi|LGPS interface is marked in (b).

Reviewer: The authors recommend the use of electronically insulating coatings for the NCM material, this would, however, after the power capabilities of the cathode electrode. Do the authors think that a coating of the sulphide particles could be an alternative pathway to decrease the degradation at the CAM|SE interface instead?

Response: Thank you for this comment. As mentioned in our previous review paper (Adv. Energy Mater. 2019, 9, 1900626.), the most ideal CAM coating layer in thiophosphate-based SSBs should meet three requisites: (1) ion-reversible, electron-blocking toward the solid electrolyte, but electron-reversible at CAM/CAM contacts, (2) conformal coverage and (3) chemically/electrochemically stable. In the present work, we emphasize that the electronic conductivity of the solid electrolyte significantly determines the degradation rate. The performance of a coating layer can be influenced by many properties (*i.e.*, electronic/ionic conductivity, mechanical ductility and electrochemical/chemical stability). In addition, the preparation method is important to control the thickness and homogeneity. In principle, a coated solid electrolyte might be alternative, but the coating should not compromise the effective ionic conductivity of the solid electrolyte itself. To the best of our knowledge there is yet no systematic study available that reports the effect of coated SE particles. This will be part of our future work.

Reviewer: To conclude, the authors should comment on and confirm the reproducibility of their study. The authors mention the use of four replicas for the capacity fading test to 3.5 V, was the dispersion of the capacity in these cells low enough to expect the same in the other cell and electrochemical tests?

Response: We agree that the reproducibility is always important in scientific studies. To avoid capacity fluctuation, we assembled the cells under the same conditions (composite cathode, LGPS separator, and In/InLi anode). Furthermore, we added ca. 12 mg composite cathode (ca. 8.4mg CAM) in different cells to ensure similar electronic/ionic percolation and voltage polarization. As we mentioned in the manuscript, we conducted the capacity calibration to remove tiny weight errors. The voltage profiles of the first cycle are provided as well in Supplementary Figure 17a, and the capacities at 3.5V are the same. For the 2nd cycle, we attribute the slightly reduced capacity to contact loss.

Supplementary Figure 17. Galvanostatic charge curves at (a) 1st and (b) 2nd cycle with different upper cutoff voltages. The capacity at 3.5 V (dash lines) is marked to evaluate the fraction of active materials. Compared with SSB charged up to 3.6 V (light blue line), the SSBs charged up to 3.7-3.9 V (yellow, green and red lines) exhibit 4.2% capacity loss.

Dr Marco Amores, Austrian Institute of Technology

Response: We thank you very much for very helpful comments; in particular the 3-electrode measurements make our results and conclusions even more reliable.

Reviewer 3

The work aims to understand the interface reaction between NCM622 and LGPS vs SOC and temperature. Systematic electrochemical impedance and ToF_SIMS measurements were used to understand the kinetics and chemistry of the interface reaction. However, I can only know how to evaluate the analysis and conclusion of the work after the following concerns are addressed.

Response: We appreciate the reviewer for spending time to review the manuscript, and we hope our responses satisfy the reviewer.

Reviewer: My main concern is whether the kinetics disclosed by Fig. 1, 2, 3 are dominated by the interface degradation of LGPS|NCM, as suggested by the authors. It seems to me some control experiments are missing here:

a) What about the voltage degradation of LGPS itself? If NCM is removed, with the cathode made only by LGPS and some conductive medium, such as VGCF used by the authors, how will the corresponding Fig. 3 look like?

Response: We thank the reviewer for the comment. As the reviewer suggested, VGCF can be used to exclude the oxygen-involving degradation. However, the electrochemical inactive nature of carbon material introduces a huge charge transfer resistance. It poses a challenge for the EIS technique.

Changes in the manuscript/actions taken: We conducted a galvanostatic intermittent titration technique (GITT) experiment with a In/InLi|LGPS|LGPS/VGCF cell (Figure R1a). 6 mg mixture ($m_{\text{LGPS}}: m_{\text{VGCF}} = 8:2$) was added to serve as working electrode. We applied a positive constant current (4 μA) to trigger the decomposition reaction. The IR drop was measured at different points. Based on Ohm's law, the resistance was plotted versus cutoff voltage (Figure R1b) and time (Figure R1c). In this case, one should note that both the potential and time are impacting the resistance increase. Although the decomposition of LGPS as such was already investigated in a previous report from our lab (Chem. Mater. 2019, 31, 8328-8337.), the potential- and time-dependence have rarely been investigated. According to the results shown below, it seems that the charging time dominates the resistance increase.

In the present work, we focus on the interfacial resistance of the LGPS|NCM interface. In combination with our electrochemical measurements and chemical characterization, electrochemical decomposition of LGPS

by delithiation and oxygen-involving degradation both take part in the interfacial degradation. In fact, these two types of interfacial reactions show different trends in resistance growth, as explained in the manuscript.

Figure R1: (a) Voltage profile of the GITT experiment with a In/InLi|LGPS|LGPS/VGCF cell. Interfacial resistance versus (b) cutoff voltage and (c) time.

Reviewer: What about the influence of the anode interface between In/InLi and LGPS? During OCV at 3.x V will the anode interface change with time and thus change the impedance?

Response: We thank the reviewer for the comment. Concerning the anode interface, we added a 3-electrode experiment to separate the resistance contribution from cathode and anode. The results prove that the resistance increase at the SE|cathode is dominating. In addition, the following EIS measurement of the Li|LGPS|In cell confirms that the In/InLi|LGPS is more stable.

Changes in the manuscript: We built the 3-electrode cell to separate the In/InLi|LGPS and LGPS|NCM622 interfaces. A tiny piece of In foil (ca. 2 mg) was embedded in the separator to serve as the reference electrode. Before cycling, the In electrode was lithiated with a weak current (30 μA) to form Li_{0.1}In. Similar to the 2-electrode cell, the 3-electrode cell was charged to 3.7 V and 3.9 V vs. In/InLi. During aging, the WE vs. CE, WE vs. RE and CE vs. RE impedance spectra were collected (Supplementary Figure 5).

With aid of the 3-electrode cell, the interfacial resistance at the SE|anode and SE|cathode interfaces can be differentiated. Compared to the anode resistance (i.e., CE vs. RE impedance), the cathode resistance (i.e., WE vs. RE impedance) is dominating the cell resistance (i.e., WE vs. CE impedance). Obviously, the In/InLi|LGPS interface is relatively stable.

Supplementary Figure 5. EIS measurements of the 3-electrode cell during resting at 25 °C. The cell was charged to 3.7 V (a-c) and 3.9 V vs. In/InLi (d-f). (a and d), (b and e) as well as (c and f) show the impedance spectra and their corresponding fitting spectra of working electrode (WE) vs. counter electrode (CE), WE vs. reference electrode (RE) and CE vs. RE, respectively. Compared to the anode|SE interfacial resistance (*i.e.*, CE vs. RE), the resistance at the SE|cathode interface (*i.e.*, WE vs. RE) increases dramatically. This result indicates that the contribution of cathode interface dominates the overall impedance. It should be noted that the introduction of RE may influence the resistance due to the thicker separator and the geometry of RE.^[2-3]

Reviewer: In fact in the charged state, since there is already Li migrated to the anode side, the anode interface stability really depends on the status of the migrated Li, e.g., if it is all alloyed with In or some of the Li is plated as Li metal without being well alloyed. As mentioned also by the authors, LGPS is known to be unstable with Li metal at least, so it is also likely that the kinetics discussed here mainly unveils the anode interface behavior instead.

Response: We thank the reviewer for the comment. The migrated Li diffuses into the In/InLi electrode, and forms more InLi phase. The first evidence is the potential of the In/InLi anode. On the basis of the 3-electrode experiment, we can confirm that V_{CE} vs. V_{RE} is about 0 V (Figure R2). This result indicates the lithiation reaction is taking place, rather than Li plating. In addition, we added a lithiation experiment with a Li|LGPS|In cell (Supplementary Figure 4). The voltage plateau at ca. 0.6 V also demonstrates the formation of the InLi phase during discharging.

Figure R2. Voltage profiles of 3-electrode cell during charging.

Supplementary Figure 4. (a) The voltage-time profile and (b) Nyquist plots of the Li|LGPS|In cell during lithiation process. The characteristic frequency (ca. 4Hz) of the In/InLi|LGPS interface is marked in (b).

Reviewer: I thus would suggest a control experiment. Use Li₄Ti₅O₁₂ mixed with LGPS as the anode, and make the anode capacity excessive wrt cathode NCM capacity, so intercalation at the anode is secured. Will the authors still see the same kinetics as described in Fig. 1-3? LTO mixed with LGPS was actually used in two of the authors' previous Nat. Energy paper ("normal type" in Table S4 of that paper), so I believe this experiment is doable by the team.

Response: We agree that LTO anode exhibits better chemical stability against LGPS. However, its high potential plateau decreases the voltage of full cells. In addition, the overlapping in the frequency ranges of the LTO|LGPS and LGPS|NCM622 interfaces may give rise to a higher fitting error (ACS Appl. Mater. Interfaces 2018, 10, 22226-22236.). To exclude the resistance change at the In/InLi|LGPS interface, we added a 3-electrode experiment to monitor the resistance change in cathode (Supplementary Figure 5b and e) and anode (Supplementary Figure 5c and f) independent – which clearly is the best proof. The above results verify that the In/InLi|LGPS interface is more stable than the LGPS|NCM622 interface.

Supplementary Figure 5. EIS measurements of the 3-electrode cell during resting at 25 °C. The cell was charged to 3.7 V (a-c) and 3.9 V vs. In/InLi (d-f). (a and d), (b and e) as well as (c and f) show the impedance spectra and their corresponding fitting spectra of WE vs. CE, WE vs. RE and CE vs. RE, respectively.

Reviewer: It's not mentioned in the Method what type of NCM622 was used, single crystal or polycrystalline? Tap density or particle size? Commercial product or synthesized?

Response: Thank you for the comment. The NCM622 material in this work is a commercial product. As shown in Supplementary Figure 2, NCM particles are polycrystalline, and their size varies in the range of 2-8 μm.

Changes in the manuscript/actions taken: We have added this information in the experimental section.

Supplementary Figure 2. SEM image of NCM622 particles.

Reviewer: It's not clear from the description in Method about the anode assembly. Is that

LGPS|In(100um)|Li(120um)?

Response: Thank you for the comment. An indium foil (0.1 mm in thickness, 8 mm in diameter) and a lithium foil (0.1 mm in thickness, 4 mm in diameter) were used.

Changes in the manuscript/actions taken: We supplemented more details in the Supplementary Information.

Reviewer: It's mentioned 12 mg composite cathode powder, but what is the cathode active material loading (mg/cm²)? From the 3:7 LGPS:622 and the anode In diameter of 8mm, it seems the authors are using a high loading of at least 16 mg/cm². This means a lot of Li will be migrated to the anode side in charge, and insufficient alloying with In together with Li metal plating might be very possible, so that the anode interface stability could be an issue.

Response: Thank you for the comment. The amount of Li in cathode and anode is calculated below: Based on the molar mass of NCM622 (96.9 g mol⁻¹), 12 mg composite cathode ($m_{\text{NCM622}}: m_{\text{LGPS}}=7: 3$) contains 86.7 μmol Li in the NCM622. We assume 72.2% Li (refers to 200 mAh g⁻¹) could be reversibly inserted/extracted from the NCM. This corresponds to 62.7 μmol Li that is transferred between cathode and anode. For the In/InLi anode, the amount of In foil (n_{In}) is 320 μmol , and the amount of Li foil (n_{Li}) is 96.5 μmol . Before cycling, the Li fraction (x) in Li_xIn is therefore $96.5/320 = 0.3$. After charging, 62.7 μmol Li atoms have been transferred from cathode to anode, and the Li fraction (x) in Li_xIn has increased to $(96.5+62.7)/320 = 0.5$. Therefore, the Li fraction (x) in Li_xIn ranges between 0.3 and 0.5 and is well in the two-phase region of the In-InLi mixture.

To check the resistance at the In/InLi interface, we built a Li|LGPS|In cell, and monitored the interfacial resistance during lithiation. As shown in the **Supplementary Figure 4a**, the first voltage plateau corresponds to the In/InLi two-phase region. The semi-circle at low frequency (ca. 4Hz) is attributed to the In/InLi|LGPS interfacial resistance. The resistance remains stable during long-term lithiation, even after the formation of the Li_3In_2 phase.

Supplementary Figure 4. (a) The voltage-time profile and (b) Nyquist plots of the Li|LGPS|In cell during lithiation process. The characteristic frequency (ca. 4Hz) of the In/InLi|LGPS interface was marked in (b).

Reviewer: What about if the authors lower the cathode loading significantly, such as by 5 to 10 times? Will the same kinetics in Fig. 1-3 still be observed?

Response: Thank you for the comment. The cathode loading determines the overall resistance, and it influences the electronic/ionic percolation in the composite cathode. A lower cathode loading will introduce a much higher error. Therefore, we believe that our current results are sufficient to confirm the reproducibility.

REVIEWERS' COMMENTS

Reviewer #1 (Remarks to the Author):

The authors have genuinely made an effort to improve the quality and robustness of their work, addressing the comments and recommendations I made in the previous round of peer review. Thus, I recommend their work for publication in Nature Communications.

Dr Marco Amores, Austrian Institute of Technology

Reviewer #3 (Remarks to the Author):

Thanks for the reply. The group obviously has strong expertise in electrochemistry, and is influential in the field of solid state battery research. I personally also enjoyed reading the group's previous publications. But that's exactly why I want to make sure all my critical concerns are lifted before I can recommend.

I still believe the lower cathode loading experiment that I suggested is important. Will the increase of impedance semicircle radius still be observed at a reduced cathode loading? If it still increases, will the speed of the increase be slowed down? Will the increase become less obvious with a more reduced loading? Quantitatively, what if you reduce the cathode loading to 70%, 50%, 30% and 10% of the current one, respectively? And why does the cathode thickness matter?

I don't think such a study is time consuming and I also don't think the "error" issue related to percolation mentioned very briefly in the authors' reply makes sense to me. Because a well-mixed composite cathode won't be easily influenced by a gradual reduction of the loading as I suggested above.

Reviewer 1

The authors have genuinely made an effort to improve the quality and robustness of their work, addressing the comments and recommendations I made in the previous round of peer review. Thus, I recommend their work for publication in Nature Communications.

Dr Marco Amores, Austrian Institute of Technology

Response: We thank the reviewer for very valuable comments in the first round of peer review.

Reviewer 3

Thanks for the reply. The group obviously has strong expertise in electrochemistry, and is influential in the field of solid state battery research. I personally also enjoyed reading the group's previous publications. But that's exactly why I want to make sure all my critical concerns are lifted before I can recommend.

Response: We thank you for the strong endorsement of our previous publications.

I still believe the lower cathode loading experiment that I suggested is important. Will the increase of impedance semicircle radius still be observed at a reduced cathode loading? If it still increases, will the speed of the increase be slowed down? Will the increase become less obvious with a more reduced loading? Quantitatively, what if you reduce the cathode loading to 70%, 50%, 30% and 10% of the current one, respectively? And why does the cathode thickness matter?

Response: Thank you for the comment. In order to evaluate the impedance change with reduced cathode loading, we rebuilt two cells with 6.6 mg and 9.1 mg composite cathodes (Supplementary Figure 6a and b). After charging up to 3.5 V vs. In/InLi, the two cells with lower cathode loading also show resistance increase during aging. The Nyquist plots were fitted with the $R(RQ)/(RQ)(RQ)Q$ equivalent circuit, as before. As shown in Figure R1d, R_{CEI} increases parabolically against aging time. Compared to the normal cathode loading (12 mg), the rate constant increases with reducing the mass loading. The rate constants of three cells with 6.6 mg, 9.1mg and 12 mg composite cathode were evaluated to be 5.7, 4.0 and 1.9 $\Omega h^{-0.5}$, respectively. This result is attributed to be the high initial interfacial resistance due to low cathode loading (*J. Phys. Chem. C* **2015**, *119*, 4612-4619.). A low cathode loading with limited contact area gives rise to a high interfacial resistance. Furthermore, the rate constants were normalized by the initial charge transfer resistance (Table R1). The normalized rate constants of three cells with 6.6 mg, 9.1mg and 12 mg composite cathode are 0.14, 0.15 and 0.17 $h^{-0.5}$, respectively. The difference may be correlated with percolation effects due to different thickness. According to the theoretical simulation (*ACS Appl. Mater. Interfaces* **2020**, *12*, 12821-12833.), the electrode thickness influences the partial electronic/ionic transport, which may further affect the diffusion-controlled interfacial degradation and the kinetics. We appreciate that the reviewer insisted to look into this subject, and we are happy to demonstrate that our analysis holds independent of the cathode loading.

Supplementary Figure 6. Comparison of resistance growth in three cells with different mass loading. (a-c) Nyquist plots of three cells after charging up to 3.5 V vs. In/InLi. (d) R_{CEI} evolution with resting time.

Table R1. Comparison of rate constants and normalized rate constants in three cells with different mass loading.

Cathode loading (mg)	R_0 (Ω)	Rate constant ($\Omega \text{ h}^{-0.5}$)	Normalized rate constant ($\text{h}^{-0.5}$)
6.6	39.8	5.7	0.14
9.1	26.6	4.0	0.15
12	11.1	1.9	0.17

I don't think such a study is time consuming and I also don't think the "error" issue related to percolation mentioned very briefly in the authors' reply makes sense to me. Because a well-mixed composite cathode won't be easily influenced by a gradual reduction of the loading as I suggested above.

Response: Thank you for the comment. As we mentioned in the first round of response letter, a low cathode loading may give rise to a high weighing error during cell processing. In addition, the contact issue will be introduced if the cathode loading is less than 6 mg. As shown in Figure R2, with a lower cathode loading, the composite cathode (black region) is not well distributed on the surface of LGPS separator (white region). Therefore, the inhomogeneous distribution of composite cathode leads to SE oxidation due to the direct contact.

Figure R2. Optical images of two exemplary cells with low cathode loading. The composite cathode (black region) is not well dispersed on the surface of LGPS separator (white region).